# Demonstrating Tumor Vascular Disrupting Activity of the Small-Molecule Dihydronaphthalene Tubulin-Binding Agent OXi6196 as a Potential Therapeutic for Cancer Treatment

**DOI:** 10.3390/cancers14174208

**Published:** 2022-08-30

**Authors:** Li Liu, Regan Schuetze, Jeni L. Gerberich, Ramona Lopez, Samuel O. Odutola, Rajendra P. Tanpure, Amanda K. Charlton-Sevcik, Justin K. Tidmore, Emily A.-S. Taylor, Payal Kapur, Hans Hammers, Mary Lynn Trawick, Kevin G. Pinney, Ralph P. Mason

**Affiliations:** 1Department of Radiology, University of Texas Southwestern Medical Center, Dallas, TX 75390, USA; 2Simmons Cancer Center, University of Texas Southwestern Medical Center, Dallas, TX 75390, USA; 3Department of Chemistry and Biochemistry, Baylor University, Waco, TX 76798, USA; 4Department of Pathology, University of Texas Southwestern Medical Center, Dallas, TX 75390, USA; 5Department of Internal Medicine, University of Texas Southwestern Medical Center, Dallas, TX 75390, USA

**Keywords:** vascular-disrupting agent (VDA), bioluminescence imaging (BLI), dihydronaphthalene, combretastatin, breast tumors, kidney tumors, lung tumors

## Abstract

**Simple Summary:**

Vascular-disrupting agents promise significant therapeutic efficacy against solid tumors by selectively damaging tumor-associated vasculature. Following administration of the phosphate prodrug OXi6197, dynamic bioluminescence imaging revealed dose-responsive vascular shutdown in MDA-MB-231 human breast tumor xenografts and for the first time murine RENCA kidney tumors. Rapid vascular disruption was observed, which continued over 24 h. Histology showed vascular congestion, hemorrhage and necrosis. Twice-weekly doses of OXi6197 caused a significant growth delay in MDA-MB-231 breast tumors and many RENCA kidney tumors growing orthotopically in mice.

**Abstract:**

The vascular disrupting activity of a promising tubulin-binding agent (OXi6196) was demonstrated in mice in MDA-MB-231 human breast tumor xenografts growing orthotopically in mammary fat pad and syngeneic RENCA kidney tumors growing orthotopically in the kidney. To enhance water solubility, OXi6196, was derivatized as its corresponding phosphate prodrug salt OXi6197, facilitating effective delivery. OXi6197 is stable in water, but rapidly releases OXi6196 in the presence of alkaline phosphatase. At low nanomolar concentrations OXi6196 caused G2/M cell cycle arrest and apoptosis in MDA-MB-231 breast cancer cells and monolayers of rapidly growing HUVECs underwent concentration-dependent changes in their morphology. Loss of the microtubule structure and increased bundling of filamentous actin into stress fibers followed by cell collapse, rounding and blebbing was observed. OXi6196 (100 nM) disrupted capillary-like endothelial networks pre-established with HUVECs on Matrigel^®^. When prodrug OXi6197 was administered to mice bearing orthotopic MDA-MB-231-luc tumors, dynamic bioluminescence imaging (BLI) revealed dose-dependent vascular shutdown with >80% signal loss within 2 h at doses ≥30 mg/kg and >90% shutdown after 6 h for doses ≥35 mg/kg, which remained depressed by at least 70% after 24 h. Twice weekly treatment with prodrug OXi6197 (20 mg/kg) caused a significant tumor growth delay, but no overall survival benefit. Similar efficacy was observed for the first time in orthotopic RENCA-luc tumors, which showed massive hemorrhage and necrosis after 24 h. Twice weekly dosing with prodrug OXi6197 (35 mg/kg) caused tumor growth delay in most orthotopic RENCA tumors. Immunohistochemistry revealed extensive necrosis, though with surviving peripheral tissues. These results demonstrate effective vascular disruption at doses comparable to the most effective vascular-disrupting agents (VDAs) suggesting opportunities for further development.

## 1. Introduction

Tumor vasculature offers an attractive target since it is directly accessible to systemic therapy. The endothelium of normal blood vessels is largely quiescent, but the invasive proliferating neovasculature of tumors is less differentiated, distinct in morphology, lacks pericyte support, exhibits increased permeability, and is more responsive to angiogenic cell signaling [1], thereby offering a unique, potentially selective, target for anticancer therapy. Moreover, vascular disruption leads to downstream ischemia, hypoxia and necrosis providing massive therapeutic amplification [2,3,4].

The original concept of vascular targeting was proposed some 30 years ago [5] and several agents entered clinical trials [3,6,7,8], but none has received FDA approval for clinical use to date. Combretastatin A-4P (CA4P) has achieved orphan drug status from the European Medicines Agency [3]. Many VDAs have been shown to effectively destroy tumor vasculature leading to massive central necrosis, but a thin rim of cells around the tumor periphery survives and continues to grow. These peripheral cells may receive their nutrients form the surrounding host vasculature and thus combination treatments have been examined including radiotherapy [9,10,11], antimetabolites (such as cytarabine) [12], antiangiogenic agents (such as bevacizumab) [4] and, recently, immunotherapy [13]. For combination treatment, the timing of each component may be crucial, since VDAs cause hypoxiation and may stimulate immune response. Indeed, the appropriate timing may generate synergistic efficacy as opposed to merely additive or even depressed response [14,15,16,17]. A clinical trial (NCT03647839) based on the benzofuran analog BNC-105P in combination with check point inhibition (Nivolumab) has just finished recruiting patients and results are awaited. 

Combretastatin recently came off patent protection and we note a dramatic increase in interest in developing analogs, particularly next generation agents and effective delivery mechanisms [18,19,20,21,22,23]. Most VDA candidates function by inhibiting microtubule formation through binding to the colchicine site on the tubulin heterodimer in endothelial cells lining tumor-associated vasculature [4,24,25]. This causes rapid morphological changes, which result in dramatically increased vessel permeability, cellular detachment, vessel occlusion and vessel wall damage. In addition to seeking effective therapy combinations, there is active search for agents which may show multi-modal activity.

A diverse range of molecules based on the combretastatin structural motif featuring a trimethoxy aryl ring linked to a second aromatic ring via an unsaturated bond, additional ring, or related molecular component has been reported [18,19,20,21]. The Pinney laboratory has spearheaded the development of several small-molecule inhibitors of tubulin polymerization, guided in part by pertinent structure activity relationship (SAR) correlations associated with combretastatin A-4 and related analogs [26,27,28,29,30,31,32]. Three particularly potent molecules (Figure 1) showed inhibition of tubulin polymerization in the low µM range in a cell free assay, cell cycle arrest in the sub-µM range and selective cytotoxicity in the pM to nM range: specifically the benzosuberene KGP18 [30,33], the indole OXi8006 [31,34] and the dihydronaphthalene OXi6196 [29]. As with the original combretastatins, these molecules are extremely hydrophobic, but derivatization as their corresponding phosphate prodrug salts yields water soluble agents that are readily administered intraperitoneally. Alkaline phosphatase (AP) is a widely expressed, membrane bound enzyme with at least four isoforms in humans: intestinal AP, liver/bone/kidney enzyme (tissue non-specific AP), placental AP and germ cell AP [35]. It has been widely used to facilitate drug delivery and is also expressed in mice [36]. An increase in AP is correlated with a number of cancers including breast and is a common marker for detecting increased bone formation in metastatic breast and prostate cancer [37,38]. Although elevated serum AP suggests a poor prognosis for renal cell carcinoma (RCC), the AP level is decreased in the kidney in RCC [39]. Nonetheless, abundant non-specific phosphatases rapidly release the active drugs in vivo (Figure 1) and we recently reported vascular disrupting activity of the phosphate prodrugs KGP265 (parent agent KGP18) [33] and OXi8007 (parent agent OXi8006) [31] in mice. 

Pinney et al. (part of the current team) reported preliminary studies regarding OXi6196 (also referred to as KGP03) and its phosphate prodrug OXi6197 (also referred to as KGP04) previously [27] with further details in a patent [40]. For OXi6196, an IC_50_ ~500 nM was determined for inhibition of tubulin polymerization utilizing a cell-free assay [27], with cytotoxicity in the low nM range (IC_50_ 2.2–4.5 nM) in human SK-OV-3 ovarian, DU-145 prostate, NCI-H460 lung cells and murine hemangioendothelioma MHEC-5T cancer cells [27,40]. Doses of OXi6197 in the range 25 to 100 mg/kg daily for 5 days significantly inhibited growth of MDA-MB-231 in mice [40]. Amali et al. also reported synthesis of a series of naphthalenes, as well as di- and tetra-hydro derivatives, including the OXi6196 structure and found very similar GI_50_ values [41]. There have been reports of non-invasive imaging of the effects of OXi6197 based on MRI, photoacoustics and color-Doppler ultrasound, all in A549 human lung tumor xenografts growing in the hind limb of nude rats [18,27]. We now report more extensive evaluation of OXi6196 and its phosphate prodrug OXi6197 in cell culture. Due to limited aqueous solubility of the parent agent OXi6196, we examined the efficacy of only the water-soluble prodrug OXi6197 to cause vascular disruption and tumor growth delay in vivo based on orthotopic human MDA-MB-231 human breast tumor xenografts in NOD SCID mice and orthotopic RENCA kidney tumors in syngeneic BALB/C mice. We demonstrated effective vascular disruption and tumor growth delay setting the stage for more extensive therapeutic investigations.

## 2. Materials and Methods

OXi6196 (2-methoxy-5-(3,4,5-trimethoxyphenyl)-7,8-dihydronaphthalen-1-ol) and prodrug OXi6197 disodium 2-methoxy-5-(3,4,5-trimethoxyphenyl)-7,8-dihydronaphthalen-1-yl phosphate) were synthesized by the Pinney lab, as described previously [27].

### 2.1. Cell Studies 

#### 2.1.1. Cell Cycle Analysis

The cell cycle profile was evaluated by flow cytometry using DNA determined by propidium iodide (PI) staining. Briefly, adherent MDA-MB-231 cells grown in 6-well culture plates at 37 °C with 5% CO_2_ were treated with OXi6196 or OXi6197 (0.5, 1, 5 and 10 nM) for 48 h. Both adherent and nonadherent cells were collected, washed twice with ice-cold PBS and fixed in 70% ethanol overnight at 4 °C. Cell pellets were suspended in PBS containing RNase (10 mg/mL) and stained with PI (6.7 mg/mL) at 4 °C in the dark for 2 h. The DNA content was determined based on propidium iodide fluorescence determined with a BD FACsCalibur (San Jose, CA, USA) [31].

#### 2.1.2. Western Blotting

MDA-MB-231 cells grown to 65–70% confluency were treated with 10 nM OXi6196 for varying times (0–90 min). Lysates were prepared by cold application of RIPA buffer, clarified via centrifugation, and treated with LDS/5% β-mercaptoethanol. Proteins were separated by SDS-PAGE on 4–12% Bis-Tris gradient gels (Invitrogen, ThermoFisher Scientific), then transferred to 0.2 μm PVDF membranes (EMD Millipore). After blocking, blots were incubated with primary antibodies (1:1000 for Cas-3, 1:2500 for GAPDH, Cell Signaling Technology), followed by secondary HRP-labeled goat anti-rabbit IgG antibodies (Jackson Immuno Research Labs, West Grove, PA USA). Blots were developed with Amersham ECL Prime chemiluminescent substrate and imaged with an ImageQuant LAS 4000 system (GE).

#### 2.1.3. Fluorescence Imaging of Endothelial Cells

HUVECs (passage two-four) were plated at 10,000 cells/glass coverslip (coated with 1% gelatin) with high growth factor-supplemented medium and incubated at 37 °C for 48 h to approximately 40% confluence. The cells were then treated with OXi6196 (10, 100, and 500 nM) from stock solutions made in DMSO (final concentration of DMSO < 0.5%) for 2 h. Cells were fixed and permeabilized with a solution of 4% paraformaldehyde and 0.5% Triton X (Sigma-Aldrich, St. Louis, MO, USA) in PBS. Microtubules were detected with mouse anti-α-tubulin antibody (Sigma-Aldrich), followed by incubation with Alexa Fluor™ 488- conjugated goat anti-mouse IgG (Invitrogen, Waltham, MA, USA), actin fibers were stained using Alexa Fluor™ 594-conjugated phalloidin (Invitrogen, Waltham, MA, USA) and 4′,6-diamidino-2-phenylindole (DAPI) was used for nuclear staining. Fluorescence images were collected with a 60× oil immersion objective on an Olympus FV 1000 confocal microscope using Olympus fluoview software (Olympus Imaging America Inc., Center Valley, PA, USA) [31].

#### 2.1.4. Endothelial Tube Disruption Assay

HUVECs (Invitrogen, Waltham, MA USA) were plated at a high concentration (124,000 cells/well) on 24-well culture plates (Corning Life Sciences, Glendale, AZ, USA) that had been coated with Matrigel^®^ (300 μL/well, 9.5 mg/mL, BD Pharmingen, San Diego, CA, USA) and maintained at 37 °C for 16 h in Human Large Vessel Endothelial Cell Basal Medium (formerly M200, Gibco/ThermoFisher Scientific, Waltham, MA, MA), supplemented with a high growth factor supplement kit (ATCC). The plates were checked to confirm a two dimensional capillary-like network had formed, and tube disruption was induced by the addition of varying concentrations of OXi6196 or colchicine (10 nM, 100 nM, and 1000 nM) for 2 h. Medium was removed and the cells were washed twice with M200. The plates were photographed (nine fields per well at 5× magnification) using a Canon Powershot A640 digital camera mounted onto an Axiovert 40 CFL inverted microscope (Zeiss, Thornwood, NY, USA) [31].

#### 2.1.5. Cancer Cell Culture

Human breast cancer cells MDA-MB-231-luc (kindly provided by Dr. Edward Graves, Stanford University) and mouse kidney cancer cells RENCA-luc were grown as described previously [33]. Cells were maintained in RPMI-1640 medium supplemented with 10% heat-inactivated FBS, 1% penicillin and streptomycin. The cells were verified to be mycoplasma free based on a MycoFluor™ Mycoplasma Detection Kit (Molecular Probes, Eugene, OR, USA).

### 2.2. Animal Models

All animal procedures were approved by the University of Texas Southwestern Medical Center (UTSW) Institutional Animal Care and Use Committee under APNs: 2012-0172, 2015-101060, 2017-102152, or 2018-102344-CORE.

MDA-MB-231-luc tumors were induced in the left upper mammary fat pad (MFP) of anesthetized female NOD SCID mice aged 6–8 weeks (bred in the UTSW colony) by direct injection of 1 × 10^6^ cells (suspended in 50–100 μL of a mixture of 75% culture medium/25% Matrigel^®^ (BD Pharmingen, San Diego, CA, USA). To generate RENCA-luc tumors, 2.5 × 10^5^ cells (in 1:1 PBS and Matrigel^®^) were injected subcutaneously (SC) in the thigh of donor male BALB/c mice. Tissue was harvested when the major axis of the donor tumor was between 0.5 and 1 cm (generally 2–4 weeks). Tissue was sectioned into 1–2 mm^3^ pieces, washed with sterile Hanks’ Balanced Salt Solution (HBSS) and implanted using forceps into the right kidney capsule of syngeneic male BALB/c mice (age 8–12 weeks from Envigo) [42]. 

Breast tumor growth was monitored by BLI and external caliper measurements (tumor size = [length × width × height] × π/6), and investigations were performed when the tumors reached a size of 5–8 mm. Orthotopic RENCA tumors were examined biweekly by BLI and treatment started when the peak emitted light intensity exceeded 10^6^ photons/s.

### 2.3. Bioluminescence Imaging (BLI)

Anesthetized mice (1 to 2% isoflurane) were imaged using an IVIS Spectrum^®^ small animal imaging system (PerkinElmer Inc., Waltham, MA, USA), as described previously [39]. For BLI, the mice were administered substrate D-luciferin (sodium salt; Gold Biotechnology, St. Louis, MO, USA) by subcutaneous injection in the fore back neck region (80 µL of 40 mg/mL solution in 0.9% saline). Images were acquired every minute over 35 min following luciferin injection to examine time course of the emitted light. The BLI signal was quantified as the maximum flux in photons/second using Living Image Software 4.2 (http://www.perkinelmer.com (accessed on 23 August 2022)). OXi6197 was dissolved in saline (5 mg/mL) and doses in the range 5 to 65 mg/kg were administered intraperitoneally (IP) for breast cancer studies and at 15 and 35 mg/kg for kidney cancer. To assess the changes in tumor perfusion, the BLI was subsequently determined as maximum signal during time course following administration of fresh luciferin at 2, 4, (or 6) and 24 h after drug administration, and in some cases additionally at 48 and 72 h.

### 2.4. Therapy

MDA-MB-231-luc cells (1 × 10^6^) were implanted in the frontal mammary fat pad of NOD SCID mice and allowed to grow until they reached 6 mm diameter (volume about 200 mm^3^). Treatment was initiated with twice weekly injections IP of 20 mg/kg OXi6197 (*n* = 5 mice) or saline (*n* = 6 mice). Tumor volume was measured using calipers each weekday as well as mouse weight. RENCA-luc tumors were implanted orthotopically in the kidney and allowed to grow until they exhibited a light emission of 10^6^ photons/s at which time they were treated twice-weekly IP with OXi6197 (35 mg/kg in saline solution (5 mg/mL) (*n* = 5)) or as saline controls (*n* = 7). BLI was performed weekly at least 48 h after a dose of OXi6197 and treatment continued until mice required sacrifice (based on excessive weight loss or tumor burden) or died. 

Growth was compared using Fisher’s Protected Least Significant Difference (PLSD) ANOVA based on tumor volume or BLI signal (as surrogate for tumor volume). For each tumor type non-parametric Kaplan Meir survival was examined based on Mantel-Cox log rank test. 

### 2.5. Histology

The tissues were fixed in 4% paraformaldehyde overnight and then processed and embedded in paraffin. The sections were cut at 5 µm, and routine H&E staining was performed. Immunohistochemical analysis was performed on representative 3–5 µm formalin-fixed paraffin-embedded (FFPE) whole tumor tissue sections. Staining for routinely used markers including Ki-67 (IR626, 1:100; Dako), CD31 (M0823, 1:50; Dako), PDL1 (CAL10, 1:200; Biocare Medical) was performed using a Dako automated system (Agilent). Slides were imaged using Roche Ventana iScanHT. Immunoreactivity was interpreted as “negative” if no or <5% tumor cells stained positive. Antigen expression was determined based on product of percentage of tumor cells staining positive and the intensity of expression. 

## 3. Results

### 3.1. Cell Culture Observations

Rapidly dividing MDA-MB-231 human breast cancer cells showed a distribution between G1 and G2/M phases of the cell cycle using flow cytometry (Appendix A, including control), but addition of ≥5 nM OXi6196 caused cell cycle arrest at the G2/M phase with significantly increased aneuploidy (Figure 2 and Appendix A). Time dependent apoptosis of MDA-MB-231 cells, demonstrated by increased cleavage of caspase-3, was observed when cells were exposed to 10 nM OXi6196 (Appendix A). Activated HUVECs, used as a model for the tumor endothelium, underwent dramatic cell morphology changes upon treatment with 10 nM OXi6196 as seen by confocal microscopy (Figure 3). The microtubule network disaggregated, and actin bundled into stress fibers. At higher OXi6196 concentrations, HUVECs, contracted, rounded, detached from the substrate with ultimate cell collapse and formation of cell surface blebs (Figure 3). Prodrug OXi6197 is less active in vitro, requiring 10 nM to achieve substantial cell cycle arrest (Appendix A), but microtubule disruption, increased phosphorylation of focal adhesion kinase (FAK) and generation of pFAK foci in HUVECs was apparent at this concentration by confocal microscopy (Appendix A).

HUVECs, cultured on Matrigel-coated 24-well plates, formed a capillary-like network (Figure 4 and Appendix A), but extensive tube disruption and cell rounding was observed with exposure to OXi6196 at 100 nM and 1000 nM concentrations. By comparison, tube disruption activity was only detected with colchicine treatment at 1000 nM.

### 3.2. MDA-MB-231 Breast Tumors 

Following subcutaneous administration of luciferin to MDA-MB-231-luc tumor bearing mice bioluminescent signal was observed reaching a maximum after about 15 to 20 min followed by gradual decline (Figure 5 and Appendix A). Following repeated administration of luciferin 2, 4–6 and 24 h later, quite similar light emission curves were observed though often with increase for control mice (Appendix A). When OXi6197 was administered IP following the baseline measurement, subsequent light emission was reduced in a dose dependent manner at each time point (Appendix A). Following 30 mg/kg emitted light was diminished to about 3% of baseline at 2 h and 1.5% after 6 h indicating progressive vascular shutdown. After 24 h there was some recovery, particularly at lower doses. Dose response is shown in Appendix A. Following 65 mg/kg OXi6197, the mouse appeared weak and died within 4 h, indicating that this potentially exceeded the maximum tolerated dose. Comparison with CA4P indicated that 35 mg/kg OXi6197 yielded similar effect to 120 mg/kg CA4P (Figure 5).

MDA-MB-231-luc tumors dosed twice weekly with OXi6197 (20 mg/kg) showed significant tumor growth delay compared with saline control tumors over the first 18 days (*p* < 0.0001, Figure 6). Within 11 days post treatment both groups had grown significantly compared with baseline, but at several time points the treated tumors were significantly smaller than the OXi6197 group (Figure 6): day 4 (*p* = 0.0116), day 11 (*p* = 0.0003), day 18 (*p* = 0.005). Growth curves are presented for individual tumors (Appendix A) and on day 28 three of five treated mice were still alive. Kaplan Meir analysis revealed a significant difference in survival (based on time of death or time to quadruple in volume from baseline). Tumor growth appeared very consistent for both control, and treated tumors.

### 3.3. RENCA Treatment

BLI of orthotopic RENCA-luc tumors showed much more rapid evolution of emitted light typically reaching a maximum within 5 min commensurate with high perfusion (Figure 7). Stable measurements were observed for untreated control mice over 24 h, while signal was significantly diminished after administration of 15 mg/kg OXi6197 and loss of signal was greater following 35 mg/kg. Very similar response was observed in relatively small or large tumors (0.97 g and 2.7 g, as determined upon sacrifice and excision). 

Gross examination of the tumor cut surface showed necrosis in the control tumor (Appendix A), which was estimated from H&E staining to be about 30% (Figure 8). In this case, 24 h after 15 mg/kg OXi6197, necrosis was very extensive with just a thin peripheral rim of viable tumor. Massive hemorrhage was observed following 35 mg/kg OXi6197 (Figure 8) and only about 5–10% viable tumor remained. Blood vessels were dilated, showed marked congestion and hemorrhage. Rim (50 to 100 µm) of viable tumor cells were observed around the blood vessels, surrounded by necrotic tissue beyond (Figure 8). By comparison RENCA tumors treated with OXi6197 showed many swollen vessels (Appendix A). Following 15 mg/kg many of these perivascular viable cells retained their ability to divide based on Ki67 staining (Figure 8). Following 35 mg/kg, Ki67 indicated much less proliferation and cells appeared to be degenerating in the H&E stained sections (Figure 8). Control RENCA showed more uniform distribution of dense tumor cells around blood vessels (Figure 8). Control RENCA-luc tumor showed high cellular density, with extensive vasculature (CD31) and high proliferation (high Ki67), but low PD-L1 based on immunohistochemistry (Appendix A).

At the time of clip removal 1 week after implantation, RENCA tumor-bearing mice showed BLI signal of about 10^5^ photons/s and they reached the threshold for initiating treatment after 2 to 3 weeks with a typical signal doubling time of 9 days (Figure 9). Mice with untreated tumors required sacrifice within 39 to 80 days of implantation, whereas treated mice (35 mg/kg) were sacrificed between 49 and 83 days, with median survival 42 versus 48 days for these small cohorts showing no significant difference in overall growth rate or survival time (Appendix A). One treated tumor grew particularly fast and if that is neglected, as an outlier, the remaining 4 treated tumors were significantly smaller than untreated controls over the first 3 weeks of treatment (*p* < 0.01) (Figure 9 and Appendix A).

## 4. Discussion

OXi6196 caused cell cycle arrest in MDA-MB-231 breast cancer cells at 5 nM and microtubule disaggregation in activated HUVECs at 10 nM. Capillary-like vascular tubes formed by cultured HUVECs were disrupted by 100 nM OXi6196. OXi6196 is hydrophobic, but delivery in vivo was achieved using the phosphate prodrug OXi6197, which offered improved water solubility and caused acute vascular shutdown within hours of administration in two distinct tumor types, as revealed by dynamic BLI. A significant tumor growth delay was observed in orthotopic MDA-MB-231 breast tumors treated twice weekly with OXi6197 (20 mg/kg) compared with saline controls. For orthotopic RENCA tumors histology confirmed major acute vascular congestion, hemorrhage and necrosis 24 h after receiving OXi6197 (15–35 mg/kg). Many RENCA tumors also showed a significant tumor growth delay with twice weekly dosing at 35 mg/kg. 

Both OXi6196 and the phosphate prodrug OXi6197 had previously been shown to exhibit cell cycle arrest and cytotoxicity in diverse cultured cells including breast (MDA-MB-231, MCF7), lung (H1299, H460), leukemia (K562), liver (HCT116), colon (KM20L2), ovarian (SKOV3) and prostate (DU-145) [27,40,41]. The only previous reported in vivo investigations appear to have been with respect to daily dosing of subcutaneous MDA-MB-231 tumor bearing SCID mice, where 25 or 50 mg/kg gave enhanced tumor growth delay over CA4P (50 mg/kg) [40]. We have now found significant tumor growth delay of orthotopic MDA-MB-231 tumors based on twice weekly dosing with 20 mg/kg (Figure 6). Acute vascular targeting activity had been observed using fluorescence microscopy in excised tumor sections post mortem. The effect of OXi6197 was previously demonstrated on subcutaneous lung tumors in nude rats using MRI, color-Doppler ultrasound and photoacoustics [18,27,43]. 

The triple negative MDA-MB-231 human breast cancer is a particularly popular model for testing novel agents and has been widely reported with respect to diverse vascular disrupting agents [18,44,45,46,47,48]. We found >90% vascular shutdown within 6 h of administering doses of OXi6197 greater than 35 mg/kg, which appears to be somewhat more active than CA4P, where similar shutdown required 100 mg/kg [47] or OXi8007, which required about 250 mg/kg [31]. Meanwhile, the benzosuberene analog KGP265 was effective at 5 mg/kg [33]. Lower doses are attractive, but ultimately it is the therapeutic window which is most important. Vascular disrupting agents have also been tested in syngeneic tumors generally examining the extremely aggressive 4T1 breast cancer in the BALB/c mouse [33,49,50,51].

Here, we chose to investigate the RENCA kidney tumor as a syngeneic model in BALB/c mice. Kidney tumors are often well vascularized and one might expect VDAs to be particularly effective. The RENCA tumor arose spontaneously in a male BALB/c mouse and is becoming increasingly popular for evaluating novel therapies [52,53,54,55,56,57]. Implantation in the sub-renal capsule is well described [58,59,60] and control mice typically live between 40 and 60 days [58]. To facilitate effective monitoring of orthotopic tumor growth and metastasis, cells have been transfected to expresses luciferase enabling BLI [61,62]. Tumors are well vascularized and perfused and we found maximum BLI intensity within about 5 min (Figure 7), which is considerably faster than for many tumor types (Figure 5 and Appendix A) and various reports in the literature [18,33,63,64]. RENCA has previously been investigated with respect to the VDAs DMXAA [54], ZD6126 [65], and BNC-105P [44], each of which entered clinical trials, as well as recent novel molecules OXi8007 [18] and KGP265 [33]. We found similar efficacy in both the orthotopic MDA-MB-231-luc and RENCA-luc tumors with about 60% loss of BLI signal following 15 mg/kg OXi6197. 

DMXAA was reported to cause massive hemorrhage and necrosis in RENCA tumors. Although no tumor growth delay was seen as a monotherapy in RENCA tumors, combination with the mTor inhibitor everolimus did yield a tumor growth delay [54]. Likewise combination of BNC-105P and everolimus caused tumor growth delay in subcutaneous RENCA tumors [44]. Combined everolimus plus BNC-105P was overall unsuccessful in a Phase 1/11 clinical trial, but genetic analysis in comparison with the CHECKMATE clinical trial showed that a subset of patients being treated for metastatic renal cell carcinoma (mRCC) exhibiting a specific 18-gene signature had significantly longer mean progression free survival (mPFS) compared with everolimus alone [66]. Vascular disruption can also be induced using photodynamic therapy, where O’Shaughnessy found that combination with immunotherapy delayed tumor progression, though neither treatment alone was effective [53]. Here, we observed a considerable tumor growth delay in 4 of 5 RENCA tumors dosed with OXi6197 alone (Appendix A).

As for the combretastatins CA4 and CA1, and previous structurally inspired analogs KGP18, OXi8006 and BNC105, phosphorylation enhanced water solubility of these hydrophobic molecules allowing effective delivery in saline IP [31,36,44,67]. Prodrug OXi6197 is stable in saline, but active drug was released rapidly by alkaline phosphatase [27]. Other approaches to deliver related agents have included serinamide salts (of amino analogs) [32,48,68], and encapsulation in cyclodextrins [69] or nanoparticles (NPs) [70,71,72], which additionally served as a reservoir providing longer term delivery. In some cases NPs have been decorated to target specific antigens [73]. Another approach was based on bioreductively activated prodrug conjugates with the goal of selectively targeting hypoxic regions in tumors [50,74,75]. 

Many diverse modalities are available for pre-clinical imaging and have been successfully applied to observe acute vascular shutdown [18]. Indeed, we have previously demonstrated the effect of OXi6197 on human lung tumor xenografts growing subcutaneously in the hind limb of nude rats using magnetic resonance imaging (MRI), multi spectral optoacoustic tomography (MSOT) and color-Doppler ultrasound (CD-US) [18,27,43]. By comparison the BLI used in the current study is particularly easy to implement and provides high throughput results, analysis and interpretation. It does of course lack the spatial resolution of 3D techniques, but effectively revealed the acute vascular shutdown in the deep seated orthotopic kidney tumors. BLI does require luciferin transfected cells, but these are widely available today. 

Histology revealed high density of closely packed cells in the control RENCA tumor with focal regions of necrosis (Figure 8 and Appendix A). Tumor cells were highly proliferative (Ki67) with focal expression of PD-L1 (15%) and preserved vasculature (Appendix A). Consistent with the imaging observations, treatment with OXi6197 led to extensive vascular congestion, vasodilation, hemorrhage, and necrosis within 24 h (Figure 8 and Appendix A). Viable cells remained around some blood vessels as confirmed by Ki67 staining. Following the higher dose of OXi6197 (35 mg/kg), hemorrhage and necrosis were far more extensive and now most of the tumor cells around blood vessels appeared to be degenerating and necrotic. Ki67 revealed minimal proliferation (Figure 8). However, some viable tumor regions remained, primarily as a thin band around the tumor periphery. These observations coincide with similar studies of previous VDAs such as ZD6126 [65] and DMXAA [54] in RENCA cells and CA4P or CA1P (OXi4503) in Caki-1 kidney tumors [76]. Both the tumor growth delay studies and histology suggest that a combined treatment will be required to achieve complete tumor control, as investigated for other VDAs previously [3,18,77]. 

Several aspects of the current study are new, noteworthy, and significant. Twice weekly dosing of MDA-MB-231 breast tumor bearing mice with 20 mg/kg OXi6197 was found to be similarly effective to the daily dosing with 25 or 50 mg/kg OXi6197 reported previously [40]. Obviously fewer doses are cheaper, logistically simpler and less invasive though optimal dosing will need further evaluation, particularly for effective combination treatments as emphasized by others [14,15,16,17,78]. We have for the first time demonstrated efficacy of OXi6197 in orthotopic RENCA kidney tumors showing acute vascular shutdown by BLI and consistent vascular occlusion confirmed by histology. Anti-CD31 staining clearly revealed extensive vascularization in control tumors, extensive vascular damage at relatively low dose (15 mg/kg), and severe vascular occlusion and swelling at higher dose, with concomitant reduction in cell proliferation. In terms of mechanism of action, we have now shown destruction of cultured capillary networks by OXi6196. Activated HUVECs underwent dramatic cell morphology changes, the microtubule network disaggregated, and actin bundled into stress fibers. Both OXi6196 and prodrug OXi6197 caused cell cycle arrest. Many of these observations align with results obtained by treatment with combretastatin A-4, and structural analogs [31,33,79] confirming flexibility for molecular designs. Indeed, the ring-locked ethylenic bond in OXi6196 and prodrug OXi6197 ensures that potential structural isomerization and inactivation cannot occur, overcoming a potential liability in the use of CA4P [80]. This may be particularly relevant when considering long term delivery approaches based on loaded polymers/nanoparticles or targeted conjugates [50,70,73].

## 5. Conclusions

We have for the first time demonstrated dose dependent vascular shutdown caused by OXi6197 in RENCA kidney tumors growing orthotopically in mice. Doses as low as 15 mg/kg caused extensive vascular congestion, hemorrhage and necrosis and biweekly dosing led to a tumor growth delay in many individual tumors. We also confirmed acute vascular disrupting effects of OXi6197 in MDA-MB-231 breast tumors in mice and found tumor growth delay based on twice weekly dosing. As with other VDAs, OXi6197 failed to fully control tumor growth, but the promising preliminary therapeutic results suggest opportunities for future investigations to develop effective combination treatment.

## Figures and Tables

**Figure 1 cancers-14-04208-f001:**
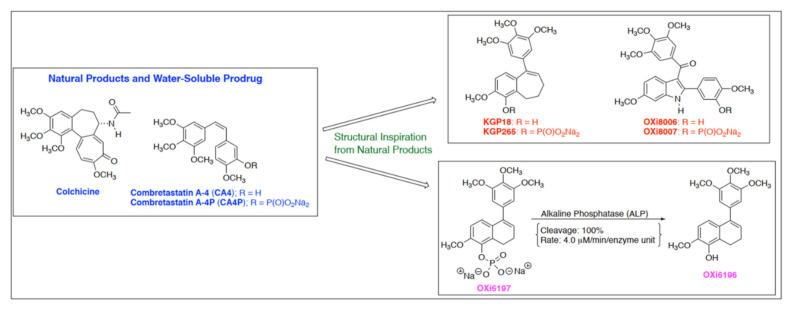
Molecular structures of OXi6196 and phosphate prodrug OXi6197, as well as related molecules exhibiting vascular disrupting activity. Exposure of prodrug OXi6197 to alkaline phosphatase leads to rapid release of OXi6196 [27].

**Figure 2 cancers-14-04208-f002:**
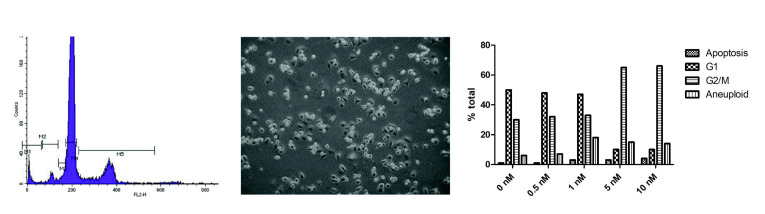
Cell cycle arrest. Concentration-dependent G2/M arrest caused by OXi6196 in MDA-MB-231 cells, as assessed by flow cytometry. G2/M blockade was observed at 5 nM with a significant increase in aneuploidy. Cells have started to round and detach. Effects on cell cycle progression and corresponding images are shown in Appendix A for OXi6196 and OXi6197, respectively.

**Figure 3 cancers-14-04208-f003:**
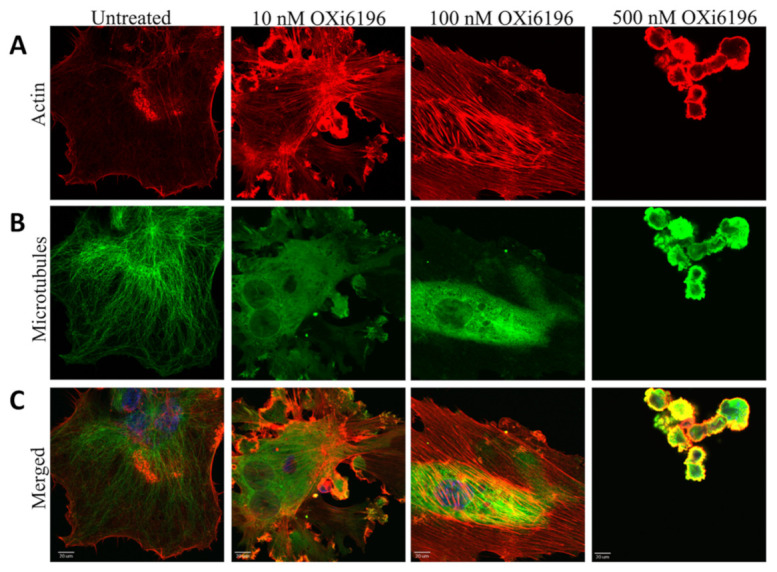
Confocal images of endothelial cells fixed and stained for actin and microtubules. Monolayers of rapidly growing HUVECs underwent concentration-dependent changes in the cell morphology with OXi6196 treatment (2 h) that demonstrated an initial loss of the microtubule structure and increased bundling of filamentous actin into stress fibers. Many cells with two nuclei were observed. At higher concentrations (500 nM) cells had detached with remaining adherent cells rounded, condensed and showing multiple blebs. Representative confocal images for endothelial cells fixed and stained with (**A**) Alexa Fluor™ 594-conjugated phalloidin (red, actin), (**B**) anti-α-tubulin antibody (green, microtubules), (**C**) merged image. Bars, 20 μm.

**Figure 4 cancers-14-04208-f004:**
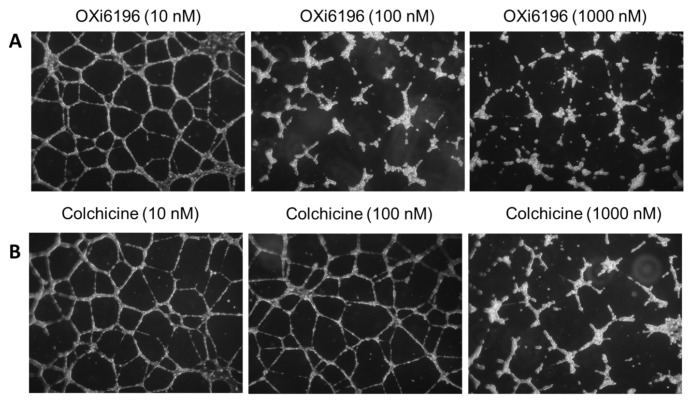
Capillary network disruption. (**A**) HUVEC capillary-like networks cultured on Matrigel^®^ showed dose-dependent tube disruption with OXi6196 treatment becoming significant at 100 nM. (**B**) Colchicine ultimately showed similar activity, but at about 10× higher concentration. Cells were imaged with an inverted microscope (5×) and nine fields were photographed per well. A minimum of three independent experiments was carried out for each treatment, and representative images are shown. An example of untreated control is shown in Appendix A.

**Figure 5 cancers-14-04208-f005:**
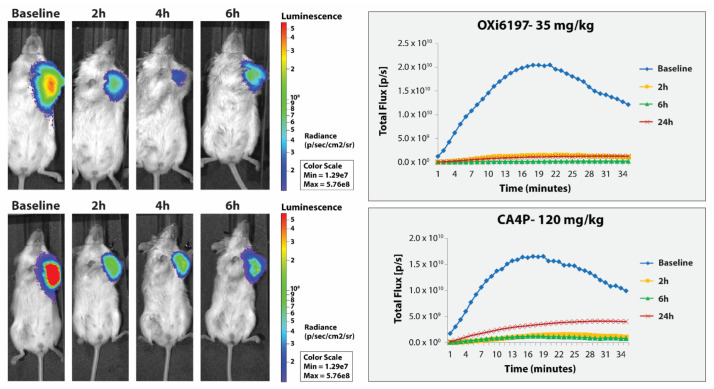
Comparison of vascular disrupting activity of OXi6197 and CA4P in orthotopic MDA-MB-231-luc tumors. (**Right**) BLI light emission curves obtained after administering luciferin to mice at various times after VDA. Significantly reduced light emission was observed following administration of VDA. (**Left**) Heat maps show light emission intensity at about 12 min after administering luciferin overlaid on photographs of mice. Dose response presented in Appendix A.

**Figure 6 cancers-14-04208-f006:**
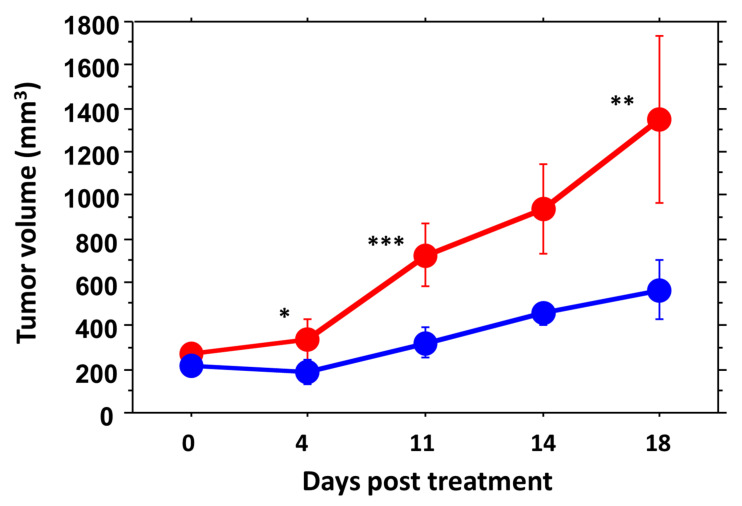
Growth curves for groups of orthotopic MDA-MB-231-luc tumors comparing saline controls (red, *n* = 6) and OXi6197 (20 mg/kg twice weekly, blue *n* = 5). At baseline the two groups had similar mean volumes, but within 4 days of treatment mean tumor volume was significantly different: 4 days (* *p* < 0.05), day 11 (*** *p* < 0.001) and day 18 (** *p* < 0.01). Curves for individual tumors are presented in Appendix A.

**Figure 7 cancers-14-04208-f007:**
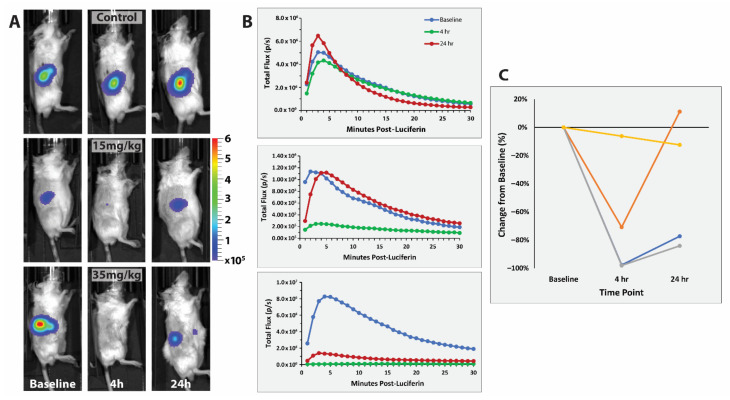
Vascular disruption assessed in RENCA-luc tumors using dynamic BLI. Effect of OXi6197 on RENCA-luc tumors growing orthotopically in the right kidney of syngeneic BALB/c mice. (**A**) Top row. Light emission observed from control tumor about 5 min after administering luciferin at baseline and 4 h and 24 h after saline. Middle row) Similar to A, but for a mouse which received OXi6197 (15 mg/kg IP). Bottom row) Similar to B, but for a mouse receiving 35 mg/kg OXi6197. (**B**) Curves showing evolution of light intensity at various times following OXi6197. Baseline (blue), 4 h (green) and 24 h (red). (**C**) Comparison of relative signal intensity indicating dose response for four tumors dosed with saline (control), 15 mg/kg and 35 mg/kg (including one small and one large: 0.97 g and 2.7 g, respectively, as determined upon sacrifice and excision).

**Figure 8 cancers-14-04208-f008:**
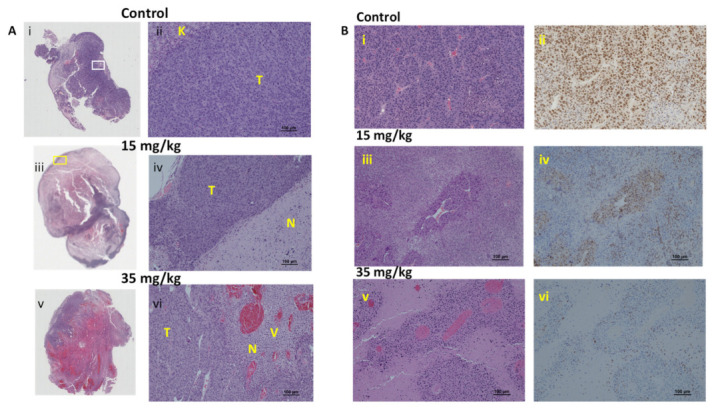
Histology of RENCA-luc tumors. (**A**) H&E stained whole mount sections and corresponding magnified regions obtained from RENCA-luc tumors 24 h after administering saline or OXi6197. (**i**) The control tumor from the mouse receiving saline shows primarily viable tissue with minimal necrosis, (**ii**) The inset shows tightly packed high-grade tumor cells (T) and remaining normal kidney from site of tumor implantation (K), (**iii**) tumor exposed to 15 mg/kg OXi6197 showing extensive necrosis (N) and thin viable rim and focal areas of hemorrhage, (**iv**) expanded view from region in yellow box showing thin rim of viable tumor (T) and extensive necrosis (N), (**v**) Section from tumor exposed to 35 mg/kg OXi6197 shows massive hemorrhage, and (**vi**) highly congested blood vessels (**v**) and extensive necrosis (N). (**B**) H&E and Ki67 stained sections from the same tumors showing findings associated with vasculature. (**i**) Control tumor shows densely packed cells and many compressed blood vessels, (**ii**) The cells show high proliferation activity, (**iii**) Following 15 mg/kg OXi6197 blood vessels appeared dilated surrounded by rim of viable cells, which (**iv**) showed retained proliferation, (**v**) Following 35 mg/kg, vessels appeared dilated and congested, and (**vi**) surrounding cells appeared less viable, as confirmed by Ki67 staining showing minimal proliferation. Further sections stained for CD31 are shown in Appendix A.

**Figure 9 cancers-14-04208-f009:**
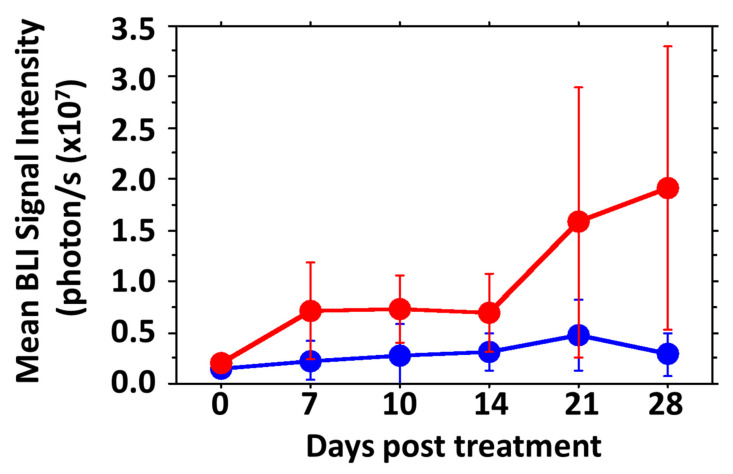
Growth curves for groups of orthotopic RENCA-luc tumors. Saline controls (red, *n* = 7) are compared with OXi6197 (35 mg/kg twice weekly, blue *n* = 4; an outlier is excluded from the group presentation). Mean tumor BLI signal as a surrogate for tumor volume was not significantly different at any given time point (*p* > 0.05), but overall the treated tumors were significantly smaller (based on light emission) than controls over the first 28 days after treatment was initiated (*p* < 0.003). Compared to baseline, control tumors were significantly larger after 21 days (*p* = 0.0176), while tumors treated with OXi6197 required 35 days to be significantly larger (*p* = 0.0035). Curves for individual tumors are presented in Appendix A.

## Data Availability

See Appendix A and further data can be shared upon request.

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
