# Peer review of "Demonstrating Tumor Vascular Disrupting Activity of the Small-Molecule Dihydronaphthalene Tubulin-Binding Agent OXi6196 as a Potential Therapeutic for Cancer Treatment"

_cancers, 2022, doi:10.3390/cancers14174208_

Round 1

Reviewer 1 Report

Comments and Suggestions for Authors are included as a .docx file. 

Reviewer 2 Report

The research article submitted by Liu et al titled “ Demonstrating Tumor Vascular Disrupting Activity of the Small-Molecule Dihydronaphthalene Tubulin-Binding Agent OXi6196 as a Potential Therapeutic for Cancer Treatment” focusses on establishing the efficacy of Oxi6196 on tumor growth via its inhibitory effect on tumor vasculature. The research study is interesting but the enthusiasm is significantly curbed due to the following issues:

Major Comments:

1.     Lack of mechanistic studies – The authors have demonstrated efficacy for Oxi6196 in inhibiting tumor angiogenesis. However, further experiments are needed to elucidate the underlying mechanism. Does Oxi6196 modulate any oncogenic signaling pathways? How does Oxi6196 modulate cytoskeletal remodeling? 

2.     The authors should consider conducting experiments delineating the effect of Oxi6196 on tumor cells vs vasculature.

3.     Overal weak sample size and lack of statistical analyses. 

Minor Comments:

1.     Figure 2 needed to be correctly labelled. Control/Vehicle group is missing in the cell cycle data. Control cell image for aneuploidy is missing. Lack of statistical presentation and analysis. 

2.     Figure 4 is missing control group. 

3.     Figure 6 lacks sample size and statistical analysis. N=2 in control group is not sufficient to demonstrate a difference with treatment.

4.     Authors should consider adding CD31 staining to demonstrate pathological reduction in tumor angiogenesis.

5.     Figure 9 lacks adequate sample size and statistical analysis.

Round 2

Reviewer 1 Report

Although the Authors included all my comments and corrections in the manuscript and answered all my questions I still have concerns about the novelty o this paper.

Reviewer 2 Report

The authors have satisfactorily addressed the concerns raised in the initial review. 

Author Response

Thank you very much for your valuable comments.